# Semantic Robustness Certification for Vision-Language Models

Peiyu Yang [1]  Paul Montague [2]  Feng Liu [1]  Andrew C. Cullen [1]  Amardeep Kaur [2]
Christopher Leckie [1]  Sarah M. Erfani [1]

## Abstract

Vision-language models (VLMs) are now widely used in downstream tasks. However, real-world applications often expose VLMs to distribution shifts induced by semantic variation (e.g., shape, size, and style). Robustness certification determines if a model's prediction changes when transformations are applied to its input. While most certification frameworks study geometric or pixel-level transformations over inputs, this work proposes a novel framework that enables certifying VLM robustness under semantic-level transformations. Leveraging the open-vocabulary capability of VLMs, we use text prompts as semantic proxies to construct transformations parameterized by an extent that controls the degree of semantic variation. By characterizing the VLM decision boundary in closed form, our framework quantitatively certifies extent intervals for which the predicted class remains unchanged under the semantic transformation. Our framework is the first to certify VLM robustness under semantic-level variations without requiring additional data for each variation, making it practical to apply. Experiments on both synthetic and real-world data show that our framework enables certifying robustness under diverse semantic variations across scenarios. Code is available at https://github.com/ypeiyu/vlm-semantic-cert.

## 1. Introduction

Vision-language models (VLMs) (Radford et al., 2021; Li et al., 2022; Alayrac et al., 2022) align images and text into a shared embedding space, enabling direct matching between vision and language for open-vocabulary reasoning.

[1]School of Computing & Information Systems, University of Melbourne, Australia [2]Defence Science and Technology Group, Australia. Correspondence to: Peiyu Yang <peiyu.yang@unimelb.edu.au>.

*Proceedings of the 43rd International Conference on Machine Learning*, Seoul, South Korea. PMLR 306, 2026. Copyright 2026 by the author(s).

This transferable interface has made VLMs a foundation for diverse downstream tasks such as detection, classification, and visual question answering (Du et al., 2022; Miyai et al., 2023; Xiao et al., 2024; Li et al., 2024). Despite their strong performance, VLM predictions can be fragile to visual semantic variations, raising concerns in high-stakes applications (Fang et al., 2022; Crabbé et al., 2023).

To provide guarantees of a model's prediction invariance under input variations, robustness certification has been widely studied (Zhang et al., 2018; Cohen et al., 2019). Given allowable transformations $\gamma$ constrained by an extent $\varphi$ that bounds the strength of $\gamma$, it aims to determine the range of $\varphi$ for which the prediction remains unchanged under transformations. Most certificates (Lecuyer et al., 2019; Cohen et al., 2019) focus on pixel-level transformation within an $L_p$ ball, but cannot capture real-world semantic variations. Other works extend certification to closed-form geometrical transforms (e.g., rotations and translations) (Balunovic et al., 2019; Li et al., 2021), but they remain limited to a small set of hand-designed transformations. Recent works perform certification in the latent space of generative models (Mirman et al., 2021; Yuan et al., 2023), enabling advanced semantic transformations (e.g., facial attributes and weather conditions). However, the requirement of substantial training data for each semantic variation limits their practical use (Wang et al., 2023b). Since semantics are highly entangled in the input space, it is challenging for existing works to formulate transformations for these variations.

In this work, we reformulate certification for the VLM embedding space, where semantics are encoded in the geometry induced by cosine similarity, which supports formalizing semantic transformations. Leveraging the open-vocabulary grounding of VLMs, we use a pair of text prompts as semantic proxies to specify the source and target semantics of a variation. We identify that this semantic variation is confined to a two-dimensional subspace spanned by the corresponding textual embeddings. Projecting an image embedding onto this subspace yields a semantic extent that quantifies the strength of the target semantic relative to the source. By varying this extent, we construct a parameterized transformation in embedding space that models the semantic variation. Figure 1 shows that, for an image of *Gyoza*, text prompts can serve as semantic proxies for diverse

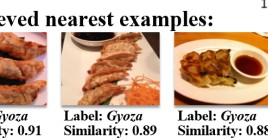

**Input image (*gyoza*):**

**Predicted prompt:**
*"A photo of a gyoza"*

**Target semantic (shape):** *triangular*
**Proxy:** *"A photo of triangular gyoza."*
**Certificates over semantic extent φ:**

| Gyoza | Samosa |
|---|---|

0.00      0.77      1.00

**Retrieved nearest examples:**

Label: *Samosa*   Label: *Samosa*   Label: *Samosa*
Similarity: 0.92   Similarity: 0.92   Similarity: 0.91

**Target semantic (style):** *soft*
**Proxy:** *"A soft photo of gyoza."*
**Certificates over semantic extent φ:**

| Gyoza | Dumplings |
|---|---|

0.00      0.63      1.00

**Retrieved nearest examples:**

Label: *Dumpling*   Label: *Dumpling*   Label: *Dumpling*
Similarity: 0.90   Similarity: 0.88   Similarity: 0.88

**Target semantic (scene):** *on a plate*
**Proxy:** *"A photo of gyoza on a plate."*
**Certificates over semantic extent φ:**

| Gyoza |
|---|

0.00      1.00

**Retrieved nearest examples:**

Label: *Gyoza*   Label: *Gyoza*   Label: *Gyoza*
Similarity: 0.91   Similarity: 0.89   Similarity: 0.88

*Figure 1.* Illustration of our semantic robustness certificates for VLMs. Each column specifies a target semantic with a text proxy. The certified prediction-invariant intervals are visualized over a normalized semantic extent $\varphi \in [0, 1]$. Nearest images are retrieved from dataset via similarity to the transformed embedding ($\varphi = 1$) as visual references, with labels and similarities shown.

variation types (e.g., shape, style, and scene), allowing the construction of semantic transformations for specific target semantics (e.g., *triangular*, *soft*, and *on a plate*).

With the established transformation, we develop a certification framework over the semantic extent. We characterize the decision boundary of VLM classifiers, where the embedding space is partitioned into Voronoi decision regions. The closed-form decision boundary allows us to analytically determine prediction changes under the transformation. Our framework produces a precise partition of the extent range into prediction-invariant intervals, with each interval annotated by its predicted class. Figure 1 visualizes prediction-invariant intervals certified over the semantic extent $\varphi$. For the target semantic *triangular*, increasing $\varphi$ strengthens the *triangular* attribute of the input *Gyoza*. The certificate shows that the prediction remains *Gyoza* under this transformation for $\varphi < 0.77$ and flips to *Samosa* beyond it. Our framework is evaluated on both generated and real-world images under semantic variation across diverse domains. Results show that our semantic transformations remain consistent with the intended semantic variation and accurately capture prediction changes along the semantic extent for VLMs. Overall, our work is the first to certify semantic robustness without requiring any additional data for each variation. This provides a practical basis for downstream applications to monitor semantic drift, diagnose failure modes under semantic variations, and characterize the evolution of a model's semantic understanding.

Our main contributions are summarized as follows.

1. We leverage text prompts as semantic proxies to formalize semantic transformations for VLMs.

2. By characterizing a VLM's decision boundary, our framework certifies precise prediction-invariant intervals.

3. Evaluations on both synthetic and real-world data show that our transformations align with the target semantics and that the certificates match prediction changes.

## 2. Related Work

**Robustness in VLMs.** Vision-language models connect visual inputs with natural-language supervision, supporting zero-shot recognition, open-vocabulary segmentation, and visual reasoning (Radford et al., 2021; Li et al., 2022; Alayrac et al., 2022; Zou et al., 2023). Despite this flexibility, VLM predictions can degrade under out-of-distribution inputs and adversarial perturbations (Schlarmann et al., 2024; Zhang et al., 2024a; Zhu et al., 2025). In response, existing work has studied VLM robustness through distribution-shift adaptation (Ming et al., 2022; Shu et al., 2023), adversarial and visual-security analysis (Zhao et al., 2023; Schlarmann et al., 2024; Li et al., 2025a; Xu et al., 2025), and multimodal optimization or distillation (Zhang et al., 2024b; Li et al., 2025b; Zhou et al., 2025). Related explanation methods connect classifier predictions to human-interpretable concepts or local evidence (Kim et al., 2018; Yang et al., 2023a;b), while recent VLM-based counterfactual methods use embedding-space semantic structure to explain classifier behavior (Kim et al., 2023). Despite known modality gaps (Liang et al., 2022), recent analyses further suggest that VLM representation spaces encode semantic structures useful for interpretation (Bhalla et al., 2024; Sonthalia et al., 2025). These studies improve robustness or interpret model behavior under observed shifts, whereas our work provides closed-form certificates of prediction-invariant intervals over a language-specified semantic extent.

**Robustness Certification.** Robustness certification aims to certify prediction invariance under a specified set of input transformations. Probabilistic approaches leverage statistical inference to provide robustness guarantees with confidence (Lecuyer et al., 2019; Cohen et al., 2019). Representative methods, including PixelDP (Lecuyer et al., 2019), randomized smoothing (Cohen et al., 2019), and TSS (Li et al., 2021), certify robustness by bounding class probabilities under randomized noise or transformations. In contrast to probabilistic guarantees, deterministic incomplete verifiers, such as AI2 (Gehr et al., 2018), DeepPoly (Singh et al., 2019), CROWN (Zhang et al., 2018), and PRIMA (Müller et al., 2022), employ convex relaxations or abstract domains

to provide sound but conservative guarantees. To mitigate the precision loss of relaxations, complete verifiers such as ReluVal (Wang et al., 2018), $\beta$-CROWN with branch and bound (Wang et al., 2021), and MN-BaB (Ferrari et al., 2022) employ iterative refinement strategies that systematically partition the search space and guarantee exactness. Building on ExactLine (Sotoudeh & Thakur, 2019), Approx-Line (Mirman et al., 2021) and GCERT (Yuan et al., 2023) certify robustness over semantic variations represented in a generative model's latent space.

**Input Transformations.** A certificate is defined with respect to allowable transformations that determine the input variations it captures. Pixel-level certificates model transformations as an $L_p$ ball to capture worst-case pixel perturbations. For explicitly parameterized transformations, methods such as DeepG (Balunovic et al., 2019) and GeoRobust (Wang et al., 2023a) model transformations with closed-form geometric parameterizations, enabling certification under affine transforms. For discrete or black-box settings, domain-specific transformations have been explored, including embedding-based transformations in DeepT (Bonaert et al., 2021) for NLP and distributional transformations in CC-Cert (Pautov et al., 2022). However, transformation models based on pixel-level perturbations or explicit geometric parameterizations cannot capture semantic-level variations that lack tractable closed-form descriptions, such as weather conditions or facial features. Consequently, methods such as ApproxLine (Mirman et al., 2021) and GCERT (Yuan et al., 2023) represent semantic transformations in the latent space of generative models, which can encode more complex semantic variations for certification. However, training such generative models requires sufficient in-domain data under the target semantic variation. In contrast, we model semantics in the multimodal embedding space of VLMs, enabling open-vocabulary semantics and supporting broad semantic coverage across domains.

## 3. Problem Statement

In this work, we focus on robustness certification for VLMs. VLMs are typically built on dual encoders that jointly learn visual and textual representations in a shared unit embedding space (Radford et al., 2021; Li et al., 2022; Alayrac et al., 2022). Let $\mathbb{S}^{d-1} := \{e \in \mathbb{R}^d : \|e\|_2 = 1\}$ denote the unit embedding sphere in VLMs. For an image $x$ and a prompt set $\{t_c : c \in \mathcal{C}\}$ over a label set $\mathcal{C}$, VLMs map $x$ and $t_c$ to a shared embedding space through a visual encoder $f_{\text{img}}$ and a textual encoder $f_{\text{text}}$, producing embeddings in $\mathbb{S}^{d-1}$. Denoting the embeddings $z := f_{\text{img}}(x)$ and $u_c := f_{\text{text}}(t_c)$, the VLM classifier $f$ acts on the shared embedding space as

$$f(z) := \arg\max_{c \in \mathcal{C}} \langle z, u_c \rangle, \qquad (1)$$

where $\langle \cdot, \cdot \rangle$ denotes the Euclidean inner product, which equals cosine similarity since all embeddings lie on $\mathbb{S}^{d-1}$.

**Our Objective.** For an input $x$ with a source semantic $a$, we formalize its variation from $a$ to a target semantic $a'$ in the embedding space using a transformation $\gamma : [\varphi_a, \varphi_{a'}] \to \mathbb{S}^{d-1}$ parameterized by an extent $\varphi \in \mathbb{R}$. Here, $\varphi_a$ denotes the source extent, and $\varphi_{a'}$ specifies the target extent to be certified. For example, in Figure 1, when the semantic variation is the *triangular* shape of a *Gyoza*, $\gamma$ formalizes the strength of the triangular attribute, and a larger $\varphi$ corresponds to a more triangular appearance of the input. Our goal is to certify whether the prediction remains invariant for all $\varphi \in [\varphi_a, \varphi_{a'}]$ under $\gamma$.

**Semantic Shift Model.** Consider admissible semantic transformations $\gamma(\varphi; z)$ of embedding $z$. We define the prediction for an input to be semantically robust over an extent interval $[\varphi_a, \varphi_{a'}]$ if it remains invariant along this range, i.e.,

$$f(\gamma(\varphi; z)) = f(z), \quad \forall \varphi \in [\varphi_a, \varphi_{a'}], \qquad (2)$$

where $\gamma(\varphi; z)$ denotes the transformed embedding at extent $\varphi$. Our framework certifies a labeled partition of the entire extent range $[\varphi_a, \varphi_{a'}]$ into subintervals $[\varphi_\ell, \varphi_{\ell+1})$ such that $f(\gamma(\varphi; z))$ is constant on each subinterval. The resulting labeled partition provides a complete robustness certificate under $\gamma$, in that each subinterval is maximal with a constant predicted label. For a fixed embedding $z$, we write $\gamma(\varphi) := \gamma(\varphi; z)$ for brevity in what follows.

## 4. Methodology

In this section, we develop our robustness certification framework by (i) characterizing semantics in the shared VLM embedding space, (ii) constructing a semantic transformation on input embeddings, and (iii) certifying semantic robustness under semantic variations.

### 4.1. Structured Semantics in the Embedding Space

We begin by characterizing how semantics are represented in the VLM embedding space, and then show that each semantic variation can be confined to an embedding subspace.

#### 4.1.1. ADDITIVE SEMANTICS IN EMBEDDING SPACE

A VLM maps an image $x$ and text $t$ to unit embeddings $z, u \in \mathbb{S}^{d-1}$ in a shared embedding space. For an input visual embedding, VLMs make predictions by comparing similarities to textual embeddings of class labels. Therefore, the prediction rule of VLMs induces a vector representation of semantics in the embedding space. Let $v_a \in \mathbb{S}^{d-1}$ denote the embedding vector corresponding to semantic $a$. Consistent with the similarity-based prediction rule of VLMs, we adopt the following assumption to quantify semantic strength in the embedding space.

**Assumption 4.1** (Similarity-based Semantic Strength). For any semantic $a$ with the embedding $v_a$, the semantic strength

of $a$ at any query embedding $e \in \mathbb{S}^{d-1}$ is measured by cosine similarity to $v_a$.

Under Assumption 4.1, for any query embedding $e \in \mathbb{S}^{d-1}$, we define its semantic strength with respect to $a$ as

$$D_a(e) := \langle e, v_a \rangle \ \in [-1, 1]. \tag{3}$$

This definition specifies the semantics in the embedding space induced by the VLM similarity geometry.

Since semantic strength is measured by cosine similarity, semantic strengths are additive under linear combinations of embeddings. We formalize this property below.

*Remark* 4.2 (Additive Semantic Strength). For any semantic embedding $v_a$, if a query embedding $e$ admits a linear decomposition over semantics $\{v_{a_i}\}$, i.e., $e = \sum_i \alpha_i v_{a_i}$, then its semantic strength with respect to $a$ decomposes additively as $D_a(e) = \sum_i \alpha_i D_a(v_{a_i})$.

This additive property of semantics shows that a semantic can be fully represented and interpreted through the linear decomposition of its embedding in VLMs. Such a property is absent in conventional neural networks whose predictions are not based on a similarity rule.

### 4.1.2. SEMANTIC PLANE

To specify a semantic variation, we consider a pair of semantics $(a, a')$ with embeddings $v_a$ and $v_{a'}$ in the shared embedding space. Assume that $v_a$ and $v_{a'}$ are linearly independent, i.e., $v_a \notin \text{span}\{v_{a'}\}$. We define the two-dimensional subspace spanned by $v_a$ and $v_{a'}$ as

$$\mathcal{P}_{a,a'} := \text{span}\{v_a, v_{a'}\} \subset \mathbb{R}^d, \tag{4}$$

where we refer to $\mathcal{P}_{a,a'}$ as the semantic plane of $(a, a')$. The plane $\mathcal{P}_{a,a'}$ isolates variations driven by $a$ and $a'$, where semantic variations are captured solely by the semantic strengths to $a$ and $a'$. Formally, we have the following remark.

*Remark* 4.3 (Semantic Plane). Let semantic embeddings $v_a, v_{a'} \in \mathbb{S}^{d-1}$ be linearly independent and let $\mathcal{P}_{a,a'} = \text{span}\{v_a, v_{a'}\}$. If an embedding $e \in \mathbb{R}^d$ varies only in the semantic strengths to $v_a$ and $v_{a'}$, then $e$ necessarily lies in the semantic plane $\mathcal{P}_{a,a'}$.

Remark 4.3 shows that if only the semantic specified by $(a, a')$ is varied, as reflected by changes in the strengths to $v_a$ and $v_{a'}$, the variation can be analyzed within the unique two-dimensional subspace $\mathcal{P}_{a,a'}$. In other words, controlling semantic strengths to $v_a$ and $v_{a'}$ confines the embedding variation to the semantic plane.

### 4.2. Semantic Transformation

In this section, we construct a transformation of the input embedding to model a semantic variation.

### 4.2.1. TEXT PROXY FOR SEMANTICS

Under contrastive training, text embeddings serve as anchors in the embedding space, encouraging matched image embeddings to align with them while separating mismatched embeddings (Radford et al., 2021; Jia et al., 2021). This contrastive objective aligns images and text in a shared embedding space, enabling language as a natural interface for specifying semantics. In contrast to text embeddings, image embeddings summarize the full image content and thus typically entangle multiple semantic factors. This motivates the use of text prompts as a proxy to specify semantics.

To specify a semantic variation, we use a pair of text prompts $(t_a, t_{a'})$ to represent the source semantic $a$ and the target semantic $a'$. We denote their embeddings by $u_a := f_{\text{text}}(t_a)$ and $u_{a'} := f_{\text{text}}(t_{a'})$, which serve as text proxies for specifying $v_a$ and $v_{a'}$. We define the semantic plane induced by this pair as $\mathcal{P}_{a,a'} := \text{span}\{u_a, u_{a'}\}$. By Remark 4.3, if an embedding varies only through its semantic strengths with respect to $u_a$ and $u_{a'}$, then the variation is confined to $\mathcal{P}_{a,a'}$. Therefore, $\mathcal{P}_{a,a'}$ captures the degrees of freedom of the variation specified by $(t_a, t_{a'})$. This reduces the specification of a semantic variation to selecting a prompt pair and analyzing the variation within the spanned plane $\mathcal{P}_{a,a'}$.

### 4.2.2. SEMANTIC TRANSFORMATION

Given the semantic plane $\mathcal{P}_{a,a'}$, we decompose an image embedding $z$ of input $x$ as

$$z = z_\| + z_\perp, \quad z_\| \in \mathcal{P}_{a,a'}, \quad z_\perp \perp \mathcal{P}_{a,a'}, \tag{5}$$

where $z_\|$ is the orthogonal projection of $z$ onto $\mathcal{P}_{a,a'}$ and $z_\perp$ is the orthogonal component. Since $u_a, u_{a'} \in \mathcal{P}_{a,a'}$ and $z_\perp \perp \mathcal{P}_{a,a'}$, the semantic strengths with respect to $(u_a, u_{a'})$ depend only on $z_\|$, i.e. $\langle z, u_a \rangle = \langle z_\|, u_a \rangle$ and $\langle z, u_{a'} \rangle = \langle z_\|, u_{a'} \rangle$. In addition, when $u_a$ and $u_{a'}$ are linearly independent, each $z_\| \in \mathcal{P}_{a,a'}$ admits a representation $z_\| = \alpha u_a + \beta u_{a'}$ for $(\alpha, \beta) \in \mathbb{R}^2$.

Equivalently, for any visual embedding $z \in \mathbb{S}^{d-1}$, we have

$$z = \underbrace{(\alpha\, u_a + \beta\, u_{a'})}_{\text{target semantic component in } \mathcal{P}_{a,a'}} + \underbrace{z_\perp}_{\text{semantics independent of } (a,a')}, \tag{6}$$

where $(\alpha, \beta)$ are the coordinates of the component in $\mathcal{P}_{a,a'}$, and $z_\perp \perp \mathcal{P}_{a,a'}$ captures the remaining semantics of $z$ that are independent of $(a, a')$.

To parameterize the semantic transformation, we establish an orthonormal basis of $\mathcal{P}_{a,a'}$ from $(u_a, u_{a'})$ as

$$e_1 := u_a, \quad e_2 := \frac{u_{a'} - \langle u_{a'}, u_a \rangle u_a}{\|u_{a'} - \langle u_{a'}, u_a \rangle u_a\|_2}, \tag{7}$$

where $(e_1, e_2)$ forms an orthonormal basis of $\mathcal{P}_{a,a'}$.

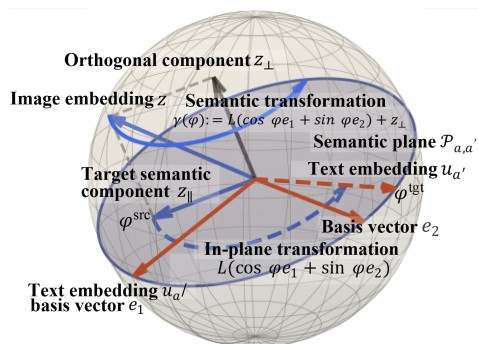

*Figure 2.* Illustration of the semantic transformation in a three-dimensional visualization of the VLM embedding space.

With the basis $(e_1, e_2)$, embeddings in $\mathcal{P}_{a,a'}$ can be parameterized by an extent $\varphi$ that controls the relative strengths with respect to $(u_a, u_{a'})$. The source semantic extent $\varphi_a \in (-\pi, \pi]$ in $\mathcal{P}_{a,a'}$ is defined from the orthogonal projection $z_\parallel$ as

$$\varphi_a := \operatorname{atan2}(\langle z_\parallel, e_2 \rangle, \langle z_\parallel, e_1 \rangle). \tag{8}$$

We assume a target extent $\varphi_{a'} \in (-\pi, \pi]$ that specifies the target semantic of the variation. We define the transformation of semantic variation over extent $\varphi \in [\varphi_a, \varphi_{a'}]$ as

$$\gamma(\varphi) := r\big(\cos\varphi\, e_1 + \sin\varphi\, e_2\big) + z_\perp, \quad \varphi \in [\varphi_a, \varphi_{a'}], \tag{9}$$

where $r := \|z_\parallel\|_2$ is the magnitude of the component of $z$ within $\mathcal{P}_{a,a'}$. The extent $\varphi$ controls this component within the plane, and $z_\perp$ is kept unchanged to preserve the semantics independent of $(a, a')$. When $\varphi = \varphi_a$, we have $\gamma(\varphi_a) = z_\parallel + z_\perp = z$. Moreover, $\gamma(\varphi) \in \mathbb{S}^{d-1}$ for all $\varphi \in [\varphi_a, \varphi_{a'}]$ since $r(\cos\varphi\, e_1 + \sin\varphi\, e_2)$ has norm $r$ for all $\varphi$ and is orthogonal to $z_\perp$.

Figure 2 illustrates an example construction of the semantic transformation in the VLM embedding space. For each extent $\varphi$, $\gamma(\varphi)$ varies only the component within $\mathcal{P}_{a,a'}$, which determines the semantic strengths with respect to text embeddings $(u_a, u_{a'})$, while keeping $z_\perp$ unchanged to preserve the remaining semantics that are independent of $(a, a')$. As $\varphi$ varies, the transformation adjusts the relative strengths to $a$ and $a'$ within $\mathcal{P}_{a,a'}$ while preserving $z_\perp$.

### 4.2.3. SEMANTIC EXTENT

The semantic transformation $\gamma(\varphi)$ is defined over an extent specified by a target extent $\varphi_{a'}$. However, semantic strength (e.g., how "big" is "big") does not admit an objective canonical scale, leaving $\varphi_{a'}$ ambiguous.

We therefore consider two practical specifications of $\varphi_{a'}$ under the *text-specified* and *image-specified* settings. In the *text-specified* setting, we anchor $\varphi_{a'}$ using the target prompt embedding $u_{a'}$ as $\varphi_{a'} := \operatorname{atan2}(\langle u_{a'}, e_2 \rangle, \langle u_{a'}, e_1 \rangle)$. This

specification treats text as a semantic reference calibrated in the VLM's similarity geometry used for prediction. In the *image-specified* setting, we anchor $\varphi_{a'}$ using a reference image $x'$ exhibiting the target semantics. Let $z' := f_{\text{img}}(x')$ and $z'_\parallel$ be its projection onto $\mathcal{P}_{a,a'}$. We set $\varphi_{a'} := \operatorname{atan2}(\langle z'_\parallel, e_2 \rangle, \langle z'_\parallel, e_1 \rangle)$. This uses direct visual evidence in the same geometry, providing a concrete specification of semantic strength when such a reference is available. These two specifications make the abstract semantic extent explicit and reproducible by yielding a precisely specified target extent, which fixes the domain of $\varphi$ over which the subsequent certification is conditioned.

### 4.3. Semantic Robustness Certification

With the established transformation of semantic variations, we develop a certification framework to determine the prediction changes under the variation.

#### 4.3.1. DECISION GEOMETRY

Consider a finite set of textual labels $\{u_c\}_{c \in \mathcal{C}} \subset \mathbb{S}^{d-1}$. Under the VLM prediction rule, each input embedding $e \in \mathbb{S}^{d-1}$ is assigned the label $f(e)$. This classification rule partitions $\mathbb{S}^{d-1}$ into Voronoi cells as

$$\mathcal{V}_c := \big\{ e \in \mathbb{S}^{d-1} : \langle e, u_c \rangle \geq \langle e, u_{c'} \rangle \ \forall c' \in \mathcal{C} \big\}, \tag{10}$$

with decision boundaries induced by pairwise bisectors

$$\mathcal{B}_{c,c'} := \big\{ e \in \mathbb{S}^{d-1} : \langle e, u_c - u_{c'} \rangle = 0 \big\}, \quad c \neq c'. \tag{11}$$

Under a semantic transformation $\gamma(\varphi)$, class flips can occur only at extents $\varphi$ where $\gamma(\varphi)$ intersects some $\mathcal{B}_{c,c'}$.

Substituting the semantic transformation $\gamma(\varphi)$ defined in Eq. (9) into the pairwise margin gives

$$m_{c,c'}(\varphi) = \langle r(\cos\varphi\, e_1 + \sin\varphi\, e_2) + z_\perp, u_c - u_{c'} \rangle$$
$$= A_{c,c'} \cos\varphi + B_{c,c'} \sin\varphi + C_{c,c'}, \tag{12}$$

where $A_{c,c'} = r\langle e_1, u_c - u_{c'} \rangle$, $B_{c,c'} = r\langle e_2, u_c - u_{c'} \rangle$, and $C_{c,c'} = \langle z_\perp, u_c - u_{c'} \rangle$. Boundary crossings along $\gamma(\varphi)$ are therefore given in closed form by solving $m_{c,c'}(\varphi) = 0$.

#### 4.3.2. CERTIFICATIONS

Our framework certifies prediction invariance over the extent by partitioning $[\varphi_a, \varphi_{a'}]$ into subintervals on which $f(\gamma(\varphi))$ remains unchanged. As $\varphi$ varies, label changes can occur only at extents where $m_{c,c'}(\varphi) = 0$ for some pair of classes $(c, c')$. By Eq. (12), each $m_{c,c'}(\varphi)$ has a closed form. Candidate change extents are therefore obtained by solving $m_{c,c'}(\varphi) = 0$ for each $(c, c')$ and retaining the solutions in $[\varphi_a, \varphi_{a'}]$. Collecting these solutions over all label pairs and sorting them yields an increasing sequence $\{\varphi_\ell\}_{\ell=0}^L$ with $\varphi_0 = \varphi_a$ and $\varphi_L = \varphi_{a'}$ that contains all extents at

which a label change can occur. This sequence induces a complete partition of the extent range into open intervals $(\varphi_\ell, \varphi_{\ell+1}) \subset [\varphi_a, \varphi_{a'}]$ on which the prediction $f(\gamma(\varphi))$ is constant. Denoting this constant label by $y_\ell := f(\gamma(\varphi))$ for any $\varphi \in (\varphi_\ell, \varphi_{\ell+1})$, the certified collection of labeled intervals is

$$\mathcal{S} := \big\{ \big((\varphi_\ell, \varphi_{\ell+1}), y_\ell\big) : \ell = 0, \dots, L-1 \big\}. \quad (13)$$

Each pair $\big((\varphi_\ell, \varphi_{\ell+1}), y_\ell\big)$ is a prediction-invariant interval under the target semantic variation, such that $f(\gamma(\varphi)) = y_\ell$ for any $\varphi \in (\varphi_\ell, \varphi_{\ell+1})$.

We further report the prediction invariance probability, which aggregates robustness over the extent range. Fix a prediction label $\hat{y} = f(\gamma(\varphi_a))$, and draw $\varphi$ uniformly from $[\varphi_a, \varphi_{a'}]$. The probability that the prediction remains $\hat{y}$ under the semantic transformation is

$$\mathbb{P}\big[f(\gamma(\varphi)) = \hat{y}\big] = \frac{1}{\varphi_{a'} - \varphi_a} \sum_{\ell : y_\ell = \hat{y}} \big(\varphi_{\ell+1} - \varphi_\ell\big), \quad (14)$$

which measures the total fraction of the semantic extent range on which the prediction is invariant.

### 4.3.3. Certificate Bounds under Misalignment

Our certificates are conditioned on the similarity-based semantic strength specification in Assumption 4.1. Consequently, cross-modal mismatch can introduce uncertainty in the resulting certificates. We model this effect via a bounded misalignment budget $\delta$ and derive conditions, parameterized by $\delta$, under which the certificates remain valid.

**Assumption 4.4** (Bounded Misalignment). *There exists $\delta \geq 0$ such that for any semantically matched pair with embeddings $z, u \in \mathbb{S}^{d-1}$ under the target semantic variation, we have $\|z - u\|_2 \leq \delta$.*

For boundary crossings $m_{c,c'}(\varphi)$, we consider any embedding $e$ within a $\delta$-neighborhood of $\gamma(\varphi)$ at each extent $\varphi$:

$$\Gamma_\delta(\varphi) := \big\{ e \in \mathbb{S}^{d-1} : \|e - \gamma(\varphi)\|_2 \leq \delta \big\}. \quad (15)$$

For any $e \in \Gamma_\delta(\varphi)$, we define $\tilde{m}_{c,c'}(\varphi; e) := \langle e, u_c - u_{c'} \rangle$. For brevity, we write $\tilde{m}_{c,c'}(\varphi)$ for $\tilde{m}_{c,c'}(\varphi; e)$. We next quantify how misalignment affects pairwise margins.

**Lemma 4.5** (Bounded Margin Gap under Misalignment). *For any $\varphi \in [\varphi_a, \varphi_{a'}]$, any $c \neq c'$, and any $e \in \Gamma_\delta(\varphi)$,*

$$\big|\tilde{m}_{c,c'}(\varphi) - m_{c,c'}(\varphi)\big| \leq \delta \|u_c - u_{c'}\|_2. \quad (16)$$

Let $\hat{y}(\varphi) := f(\gamma(\varphi))$. We derive a stability condition ensuring that $f(e) = \hat{y}(\varphi)$ for all $e \in \Gamma_\delta(\varphi)$.

**Proposition 4.6** (Stability under Misalignment). *Under Assumption 4.4, fix any $\varphi \in [\varphi_a, \varphi_{a'}]$ and let $\hat{y}(\varphi) := f(\gamma(\varphi))$. If $m_{\hat{y}(\varphi),c'}(\varphi) > \delta \|u_{\hat{y}(\varphi)} - u_{c'}\|_2$ for all $c' \neq \hat{y}(\varphi)$, then $f(e) = \hat{y}(\varphi)$ holds for all $e \in \Gamma_\delta(\varphi)$.*

Using the closed-form margin $m_{c,c'}(\varphi)$, we further localize the extents where a boundary crossing may occur under misalignment by defining the uncertainty set

$$\mathcal{U}_{c,c'}(\delta) := \big\{ \varphi \in [\varphi_a, \varphi_{a'}] : |m_{c,c'}(\varphi)| \leq \varepsilon_{c,c'} \big\}, \quad (17)$$

where $\varepsilon_{c,c'} := \delta \|u_c - u_{c'}\|_2$ is the margin tolerance. We next localize boundary-crossing uncertainty to $\mathcal{U}_{c,c'}(\delta)$.

**Lemma 4.7** (Crossing Localization). *For any $\varphi \notin \mathcal{U}_{c,c'}(\delta)$ and any $e \in \Gamma_\delta(\varphi)$, $m_{c,c'}(\varphi)$ and $\tilde{m}_{c,c'}(\varphi)$ have the same sign. Consequently, any extent $\varphi$ satisfying $\tilde{m}_{c,c'}(\varphi) = 0$ must lie in $\mathcal{U}_{c,c'}(\delta)$.*

Using the cosine form of $m_{c,c'}(\varphi)$, $\mathcal{U}_{c,c'}(\delta)$ is given by

$$\big| R_{c,c'} \cos(\varphi - \psi_{c,c'}) + C_{c,c'} \big| \leq \varepsilon_{c,c'}, \quad (18)$$

where the amplitude $R_{c,c'} := (A_{c,c'}^2 + B_{c,c'}^2)^{1/2}$ and the phase $\psi_{c,c'} := \mathrm{atan2}(B_{c,c'}, A_{c,c'})$. The boundaries solve $R_{c,c'} \cos(\varphi - \psi_{c,c'}) = -C_{c,c'} \pm \varepsilon_{c,c'}$ for $\varphi \in [\varphi_a, \varphi_{a'}]$.

Thus, for a given misalignment budget $\delta$, we identify the extents that are guaranteed prediction-invariant within $\Gamma_\delta(\varphi)$ and localize the remaining uncertainty to $\mathcal{U}_{c,c'}(\delta)$.

## 5. Experiments

In this section, we evaluate our framework on both controlled synthetic and real-world semantic variations. Experiments are conducted on publicly available CLIP VLMs (Radford et al., 2021). Before presenting the results, we describe the experimental setup below.

**Prompts for Semantics.** We use text prompts to specify the semantic variations. The prompt pair $(t_a, t_{a'})$ specifies which semantic factors are allowed to vary under the transformation. If the prompts differ beyond the intended attribute, then $\mathcal{P}_{a,a'}$ may capture additional semantic factors, and moving within $\mathcal{P}_{a,a'}$ can leak into unintended semantic variations. In our experiments, we control this effect by using prompt pairs that share the same template and content words, differing only in the attribute token. For each attribute type, we construct a comprehensive list of attribute descriptors. Multiple attribute types are considered to evaluate the generality of our framework. Figure 4 provides examples of the descriptors used for each attribute type.

**Baselines.** We evaluate against ExactLine, a complete certification method that certifies predictions over the linear interpolation path between two endpoint images (Sotoudeh & Thakur, 2019). ExactLine identifies all decision boundary crossing points along the interpolation path, yielding a complete partition of the range into prediction-invariant intervals. Our work targets semantic transformations in VLMs without requiring auxiliary inputs. Under this setting, ExactLine is the only existing framework we are aware of that

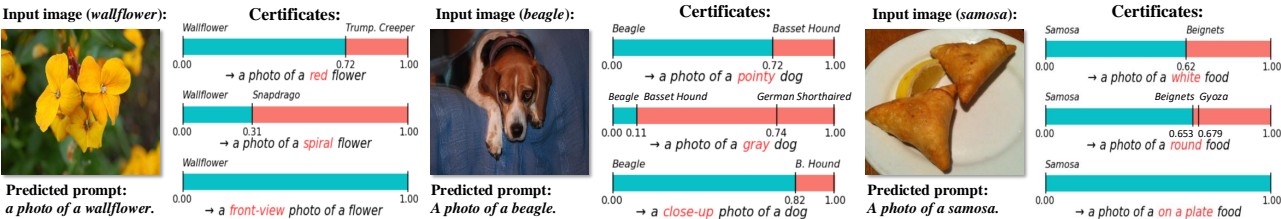

*Figure 3.* Illustration of our VLM robustness certificates. Prediction-invariant intervals are completely certified over a normalized semantic extent $\varphi \in [0, 1]$ for diverse semantic variations across domains. Text prompts serve as proxies for specifying different target semantics.

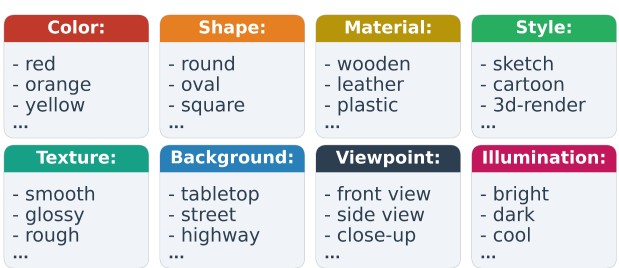

*Figure 4.* Illustration of descriptors grouped by attribute type. Using a fixed prompt template (e.g., "a photo of a {attribute} class"), we vary only the descriptor to model semantic variations.

provides complete certification without additional inputs or supervision (Mirman et al., 2021; Yuan et al., 2023), and therefore serves as our primary baseline.

**Visual Reference Transformation and Metric.** Ground-truth semantic variations are difficult to define and annotate, which makes it challenging to directly evaluate whether a specified transformation follows the intended change. To obtain a visual reference, we assume access to an image sequence $\{x_k\}_{k=0}^K$ that exhibits the target semantic variation from $a$ to $a'$. Let $z_k := f_{\text{img}}(x_k) \in \mathbb{S}^{d-1}$ be the corresponding normalized embeddings, with $z_0$ as the starting embedding. We fit a visual reference transformation as a great circle arc on the unit sphere, $\gamma^{\text{ref}} : [\varphi_a, \varphi_{a'}] \to \mathbb{S}^{d-1}$, by least squares over the samples, minimizing $\sum_{k=0}^K \|\gamma^{\text{ref}}(\varphi_k) - z_k\|_2^2$ subject to $\gamma^{\text{ref}}(\varphi_a) = z_0$. We then quantify alignment between the specified semantic transformation $\gamma(\varphi)$ and the visual reference transformation $\gamma^{\text{ref}}(\varphi)$ using the discrepancy in their prediction invariance probabilities as defined in Eq. (14). Concretely, we fix a reference label $\hat{y} = f(\gamma(\varphi_a))$ and compute the prediction invariance probability for each transformation. For each image $x_i$, let $\gamma_i^{\text{ref}}(\varphi)$ be the fitted visual reference transformation and $\gamma_i(\varphi)$ be the specified semantic transformation, and we report the mean absolute discrepancy over all $N$ images as $\frac{1}{N} \sum_{i=1}^N \left| \mathbb{P}[f(\gamma_i^{\text{ref}}(\varphi)) = \hat{y}] - \mathbb{P}[f(\gamma_i(\varphi)) = \hat{y}] \right|$.

## 5.1. Qualitative Evaluation of Semantic Variations

Since semantic variation lacks a standard quantitative ground truth, qualitative analysis plays a necessary role in validating that the evaluation reflects the intended semantic

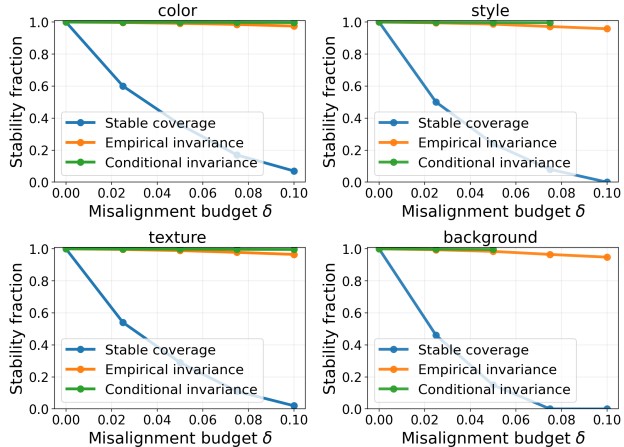

*Figure 5.* Evaluation of certificate bounds on an ImageNet subset. We sweep the misalignment budget $\delta$ for semantic variations driven by color, style, texture, and background. Stable coverage is the fraction of extents that are certified prediction-invariant. Empirical invariance is the observed fraction of invariant extents under sampled perturbations within the $\delta$-neighborhood. Conditional invariance reports empirical invariance restricted to certified extents.

change. Before presenting quantitative results, we therefore provide a qualitative study. In Figure 3, we model different semantic variations by using text prompts and present certificates of prediction-invariant intervals over the normalized extent. The figure shows that our method can track diverse semantic variations and can identify plausible flipped classes under transformations. For instance, under the target semantic *round* for the input *Samosa*, in contrast to the example in Figure 1, the prediction flips back to *Gyoza*. This suggests qualitative alignment between our semantic transformation and real-world semantic variations.

## 5.2. Evaluation of Certificate Bounds

We evaluate the certificate bounds under semantic misalignment during certification. Since estimating real-world misalignment would require additional annotation and calibration beyond the scope of this work, we evaluate user-specified misalignment budgets. For each semantic variation, we sweep the misalignment budget $\delta$ and sample perturbations within the corresponding $\delta$-neighborhood at each extent, then measure whether the prediction matches the nominal prediction along the transformation and aggre-

*Table 1.* Comparison of mean absolute discrepancy among ExactLine, our text-specified transformation (T-Spec), and our image-specified transformation (I-Spec) under synthetic semantic variations, covering both in-domain (ID) attributes and out-of-domain (OOD) attributes.

| *(a)* Oxford Pets. | | | |
|---|---|---|---|
| | ID | | OOD |
| | Color | Background | Texture |
| ExactLine | 12.6% | 10.2% | 18.1% |
| T-Spec | 6.9% | 7.3% | 7.4% |
| I-Spec | 4.1% | 3.2% | 6.7% |

| *(b)* Flowers102. | | | | |
|---|---|---|---|---|
| | ID | | OOD | |
| | Color | View | Color | Shape |
| ExactLine | 8.4% | 12.6% | 10.1% | 19.3% |
| T-Spec | 6.5% | 8.2% | 9.4% | 8.7% |
| I-Spec | 2.3% | 4.0% | 7.5% | 6.6% |

| *(c)* Food101. | | | |
|---|---|---|---|
| | ID | | OOD |
| | View | Background | Color |
| ExactLine | 14.6% | 10.7% | 9.3% |
| T-Spec | 0.0% | 6.2% | 9.6% |
| I-Spec | 0.0% | 3.9% | 6.8% |

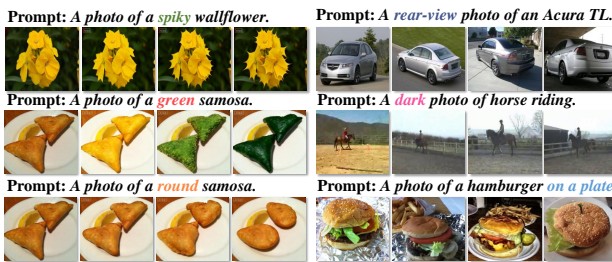

**Prompt:** *A photo of a spiky wallflower.*
**Prompt:** *A rear-view photo of an Acura TL.*
**Prompt:** *A photo of a green samosa.*
**Prompt:** *A dark photo of horse riding.*
**Prompt:** *A photo of a round samosa.*
**Prompt:** *A photo of a hamburger on a plate.*

**(a) Synthetic semantic variations.** **(b) Real-world semantic variations.**

*Figure 6.* Images of synthetic and real-world semantic variations.

gate over the extent range. Figure 5 reports stable coverage, empirical invariance, and conditional invariance. Across attributes, conditional invariance remains close to one as $\delta$ increases, supporting the validity of the bound. As expected, stable coverage decreases with larger $\delta$ because certifying invariance under stronger misalignment requires larger pairwise margins. Empirical invariance stays high over the same range, suggesting that the bound is conservative yet reliable.

### 5.3. Evaluation on Synthetic Semantic Variations

To evaluate certificates under controlled semantic variations, we use multimodal LLMs (Achiam et al., 2023; Guo et al., 2025) to generate a sequence of images exhibiting gradual semantic variations for three image recognition datasets, OxfordPets (Parkhi et al., 2012), Flowers102 (Nilsback & Zisserman, 2008), and Food101 (Bossard et al., 2014). For each dataset, we sample images across multiple categories as source images for generation. Figure 6(a) shows example image sequences generated under different semantic variations. We consider both in-domain (ID) variations that occur in the dataset and out-of-distribution (OOD) variations that do not occur in the dataset. For ExactLine and our image-based method, we set the last image from the collected images as the reference image to specify the target semantic extent. Table 1 reports the mean absolute discrepancy over the semantic extent for ExactLine and our framework. The results show that our method consistently yields lower discrepancy, indicating that the proposed transformation aligns better with the intended semantic variation. ExactLine models variation by linear interpolation between two endpoint images, which can approximate simple semantic variations such as color. However, for more complex

variations, such as viewpoint or shape, it often introduces unintended changes that are not part of the target semantics. In contrast, our transformation leads to more stable alignment across both in-domain and out-of-distribution variations.

### 5.4. Evaluation on Real-World Semantic Variations

To evaluate certificates on real-world semantic variations, we use eight image recognition datasets, including Caltech101 (Fei-Fei et al., 2004), OxfordPets (Parkhi et al., 2012), StanfordCars (Krause et al., 2013), Flowers102 (Nilsback & Zisserman, 2008), Food101 (Bossard et al., 2014), UCF101 (Soomro et al., 2012), DTD (Cimpoi et al., 2014) and FGVCAircraft (Maji et al., 2013). For each dataset, we identify the main attribute types that naturally vary within the dataset. We then construct a real-world image sequence of semantic variations, as illustrated in Figure 6(b). Compared to generated sequences, real-world sequences exhibit greater uncontrolled variation and may not isolate a target variation. To mitigate this problem, we use the VLM to rank images by their similarity to the corresponding prompt within a semantic family, thereby ordering the sequence primarily by the intended semantic. This VLM-induced ordering provides a practical visual reference for evaluating alignment. Table 2 reports the mean absolute discrepancy over the semantic extent, showing that our method consistently outperforms ExactLine across datasets and semantics, indicating alignment with real-world semantic variation.

## 6. Discussion

Beyond the formal certification guarantee, our framework offers practical benefits for analyzing, comparing, and improving VLM robustness. Certifying prediction-invariant intervals along the semantic extent characterizes where the prediction remains stable. This makes the certificate not only a robustness statement, but also a diagnostic description of the model's semantic decision geometry. Since reference images can be mapped onto the same semantic extent, the certified intervals can be related to concrete visual examples rather than only to abstract embedding coordinates. The closed-form margin further indicates proximity to semantic decision boundaries and quantifies how the certified intervals change under bounded cross-modal misalignment.

*Table 2.* Comparison of mean absolute discrepancy among ExactLine, our text-specified transformation (T-Spec), and our image-specified transformation (I-Spec) under real-world semantic variations (lower is better).

*(a)* DTD.

| | Color | Size | Texture | Illumination (ILL) |
|---|---|---|---|---|
| ExactLine | 29.9% | 18.5% | 8.7% | 11.9% |
| T-Spec | 21.3% | 10.1% | 2.4% | 0.0% |
| I-Spec | 19.3% | 8.1% | 3.9% | 0.0% |

*(b)* FGVCAircraft.

| | Viewpoint (VP) | Background (BG) |
|---|---|---|
| ExactLine | 31.9% | 13.0% |
| T-Spec | 26.8% | 14.9% |
| I-Spec | 27.4% | 11.0% |

*(c)* Caltech101.

| | Color | VP | Style |
|---|---|---|---|
| ExactLine | 2.7% | 0.0% | 10.5% |
| T-Spec | 0.0% | 0.0% | 11.9% |
| I-Spec | 0.0% | 0.0% | 7.8% |

*(d)* StanfordCars.

| | Color | VP | BG |
|---|---|---|---|
| ExactLine | 32.8% | 31.7% | 21.4% |
| T-Spec | 11.3% | 8.2% | 3.0% |
| I-Spec | 6.2% | 5.8% | 4.5% |

*(e)* Flowers102.

| | Color | Shape | VP |
|---|---|---|---|
| ExactLine | 14.9% | 23.5% | 0.0% |
| T-Spec | 11.0% | 10.3% | 0.0% |
| I-Spec | 6.2% | 8.7% | 0.0% |

*(f)* OxfordPets.

| | Texture | VP | BG |
|---|---|---|---|
| ExactLine | 5.2% | 10.3% | 11.5% |
| T-Spec | 2.9% | 0.0% | 3.2% |
| I-Spec | 0.5% | 0.0% | 0.2% |

*(g)* Food101.

| | Shape | BG | ILL |
|---|---|---|---|
| ExactLine | 14.7% | 11.2% | 7.4% |
| T-Spec | 0.0% | 0.0% | 6.8% |
| I-Spec | 0.0% | 0.0% | 5.1% |

*(h)* UCF101.

| | VP | BG | ILL |
|---|---|---|---|
| ExactLine | 30.1% | 13.6% | 7.9% |
| T-Spec | 14.1% | 0.0% | 7.3% |
| I-Spec | 4.0% | 0.0% | 2.9% |

These properties support practical uses in robustness auditing, prompt learning, and model adaptation. The certified intervals and class transitions reveal which target semantic variations induce prediction changes and support comparisons across datasets and VLMs (Fang et al., 2022; Yang et al., 2026; Jiang et al., 2025; Yu et al., 2026). For prompt engineering, interval length can serve as a certificate-aware criterion that favors stable predictions over target semantic variations, complementing performance-driven objectives (Zhou et al., 2022b;a; Jiang et al., 2024). Since the certified object is the shared image-text scoring mechanism, the same analysis can also inform downstream pipelines that reuse this score, including image-text retrieval, detection, and segmentation (Du et al., 2022; Zou et al., 2023).

The scope of the certificate is subject to two qualifications. First, the framework depends on the quality of the language proxy and the alignment between visual and textual embeddings. Our bounded misalignment analysis makes this dependence explicit and localizes uncertainty due to cross-modal mismatch, but the resulting bound can become conservative when the modality gap is large. Recent work has begun to analyze and reduce such gaps in VLM representation spaces (Liang et al., 2022; Bhalla et al., 2024), while integrating modality-gap correction into semantic certification remains an important future direction. Second, validating semantic transformations remains challenging because semantic variation is difficult to isolate in the input space. The real-world sequences used in our experiments provide practical visual references, but can still contain non-target semantic changes. The synthetic sequences offer more control over the intended progression, but can introduce artifacts from the source identity. In addition, zero

discrepancy values can occur when the visual reference path itself does not induce a prediction change. Such cases still test whether a transformation avoids spurious class transitions, but they provide limited evidence about boundary localization. These challenges motivate semantic-variation benchmarks with stronger control over target attributes and evaluation protocols that distinguish transformation alignment from the absence of prediction changes.

## Acknowledgments

The authors would like to thank Neil Marchant for his valuable input during the course of this research project. This work was supported by the Australian Defence Science and Technology (DST) Group via the Advanced Strategic Capabilities Accelerator (ASCA) program.

## Impact Statement

This work develops robustness certificates for vision-language models under target semantic variations. By characterizing how predictions evolve along a parameterized semantic extent and identifying prediction-invariant intervals, the framework supports transparent evaluation and monitoring of semantic drift, and helps localize where predictions are reliably stable under the specified semantic change. We certify prediction invariance under semantic variations specified by text prompts in a similarity space, while a bounded misalignment budget models the remaining cross-modal mismatch. Accordingly, the results are intended to complement application-specific testing and should not be interpreted as guarantees under arbitrary real-world transformations or for safety-critical deployment.

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

## A. Proof

In this section, we provide the proof of Lemmas 4.5-4.7. We begin with the proof of Lemma 4.5.

*Proof of Lemma 4.5.* Fix any $\varphi \in [\varphi_a, \varphi_{a'}]$, any $c \neq c'$, and any $e \in \Gamma_\delta(\varphi)$. Recall that the nominal pairwise margin along the path $\gamma$ is defined as

$$m_{c,c'}(\varphi) := \langle \gamma(\varphi),\, u_c - u_{c'} \rangle,$$

and the perturbed margin at extent $\varphi$ under an embedding $e$ is

$$\tilde{m}_{c,c'}(\varphi; e) := \langle e,\, u_c - u_{c'} \rangle.$$

Taking the difference and using bilinearity of the inner product yields

$$\tilde{m}_{c,c'}(\varphi; e) - m_{c,c'}(\varphi) = \langle e - \gamma(\varphi),\, u_c - u_{c'} \rangle.$$

Applying the Cauchy–Schwarz inequality gives

$$\left| \tilde{m}_{c,c'}(\varphi; e) - m_{c,c'}(\varphi) \right| \leq \|e - \gamma(\varphi)\|_2\, \|u_c - u_{c'}\|_2.$$

Finally, since $e \in \Gamma_\delta(\varphi)$ and

$$\Gamma_\delta(\varphi) = \left\{ e \in \mathbb{S}^{d-1} :\ \|e - \gamma(\varphi)\|_2 \leq \delta \right\},$$

we have $\|e - \gamma(\varphi)\|_2 \leq \delta$. Substituting this bound into the above inequality yields

$$\left| \tilde{m}_{c,c'}(\varphi; e) - m_{c,c'}(\varphi) \right| \leq \delta\, \|u_c - u_{c'}\|_2,$$

which completes the proof. $\qquad\square$

We then prove Proposition 4.6.

*Proof of Proposition 4.6.* Under Assumption 4.4, fix any $\varphi \in [\varphi_a, \varphi_{a'}]$ and let $\hat{y}(\varphi) := f(\gamma(\varphi))$. Take an arbitrary $e \in \Gamma_\delta(\varphi)$. For any $c' \neq \hat{y}(\varphi)$, define the perturbed margin

$$\tilde{m}_{\hat{y}(\varphi),c'}(\varphi; e) := \langle e,\, u_{\hat{y}(\varphi)} - u_{c'} \rangle.$$

By Lemma 4.5 applied to the pair $(\hat{y}(\varphi), c')$,

$$\left| \tilde{m}_{\hat{y}(\varphi),c'}(\varphi; e) - m_{\hat{y}(\varphi),c'}(\varphi) \right| \leq \delta\, \|u_{\hat{y}(\varphi)} - u_{c'}\|_2.$$

Hence,

$$\tilde{m}_{\hat{y}(\varphi),c'}(\varphi; e) \geq m_{\hat{y}(\varphi),c'}(\varphi) - \delta\, \|u_{\hat{y}(\varphi)} - u_{c'}\|_2.$$

By the condition of the proposition, the right-hand side is strictly positive for every $c' \neq \hat{y}(\varphi)$. Therefore,

$$\langle e,\, u_{\hat{y}(\varphi)} - u_{c'} \rangle > 0 \quad \forall c' \neq \hat{y}(\varphi),$$

equivalently,

$$\langle e,\, u_{\hat{y}(\varphi)} \rangle > \langle e,\, u_{c'} \rangle \quad \forall c' \neq \hat{y}(\varphi).$$

Thus $\hat{y}(\varphi)$ uniquely maximizes $\langle e, u_c \rangle$ over $c \in \mathcal{C}$, and by the definition $f(e) := \arg\max_{c \in \mathcal{C}} \langle e, u_c \rangle$ in (1), we conclude $f(e) = \hat{y}(\varphi)$. Since $e \in \Gamma_\delta(\varphi)$ was arbitrary, the claim holds for all $e \in \Gamma_\delta(\varphi)$. $\qquad\square$

Below, we provide the proof of Lemma 4.7.

*Proof of Lemma 4.7.* Fix any $c, c' \in \mathcal{C}$ with $c \neq c'$. Take any $\varphi \in [\varphi_a, \varphi_{a'}]$ and any $e \in \Gamma_\delta(\varphi)$. By definition,

$$m_{c,c'}(\varphi) = \langle \gamma(\varphi),\, u_c - u_{c'} \rangle, \qquad \tilde{m}_{c,c'}(\varphi; e) = \langle e,\, u_c - u_{c'} \rangle.$$

Applying Lemma 4.5 yields

$$\left| \tilde{m}_{c,c'}(\varphi; e) - m_{c,c'}(\varphi) \right| \leq \delta \left\| u_c - u_{c'} \right\|_2 \; = \; \varepsilon_{c,c'}.$$

Now assume $\varphi \notin \mathcal{U}_{c,c'}(\delta)$. By the definition of $\mathcal{U}_{c,c'}(\delta)$ in (17), we have $|m_{c,c'}(\varphi)| > \varepsilon_{c,c'}$. We show that $m_{c,c'}(\varphi)$ and $\tilde{m}_{c,c'}(\varphi; e)$ must have the same sign.

If $m_{c,c'}(\varphi) > \varepsilon_{c,c'}$, then

$$\tilde{m}_{c,c'}(\varphi; e) \geq m_{c,c'}(\varphi) - \left| \tilde{m}_{c,c'}(\varphi; e) - m_{c,c'}(\varphi) \right| \geq m_{c,c'}(\varphi) - \varepsilon_{c,c'} > 0.$$

If $m_{c,c'}(\varphi) < -\varepsilon_{c,c'}$, then

$$\tilde{m}_{c,c'}(\varphi; e) \leq m_{c,c'}(\varphi) + \left| \tilde{m}_{c,c'}(\varphi; e) - m_{c,c'}(\varphi) \right| \leq m_{c,c'}(\varphi) + \varepsilon_{c,c'} < 0.$$

In either case, $\tilde{m}_{c,c'}(\varphi; e)$ has the same sign as $m_{c,c'}(\varphi)$. Since $e \in \Gamma_\delta(\varphi)$ was arbitrary, this proves the first claim. For the second claim, suppose there exist $\varphi \in [\varphi_a, \varphi_{a'}]$ and $e \in \Gamma_\delta(\varphi)$ such that $\tilde{m}_{c,c'}(\varphi; e) = 0$. If $\varphi \notin \mathcal{U}_{c,c'}(\delta)$, then the first part implies that $\tilde{m}_{c,c'}(\varphi; e)$ is strictly positive when $m_{c,c'}(\varphi) > 0$, and strictly negative when $m_{c,c'}(\varphi) < 0$. In either case, $\tilde{m}_{c,c'}(\varphi; e) \neq 0$, contradicting the assumption that $\tilde{m}_{c,c'}(\varphi; e) = 0$. Therefore, $\varphi \in \mathcal{U}_{c,c'}(\delta)$. $\square$

## B. Experimental Setup

In this work, we use the publicly available pretrained CLIP ViT-B/16 model released by OpenAI (Radford et al., 2021). All experiments are conducted using an NVIDIA 3090Ti GPU (24GB), a 16-core 3.9GHz Intel Core i9-12900K CPU, and 128GB RAM.

To evaluate semantic robustness under controllable semantic extents, we construct two complementary evaluation sets: (i) *real-world* semantic-variation sequences collected from existing datasets, and (ii) *synthetic* sequences that explicitly instantiate both ID and OOD semantic variations. We emphasize that building a fully controlled semantic-variation benchmark remains practically challenging, as the semantic edits must preserve class identity while avoiding spurious artifacts that can confound robustness analysis.

**Synthetic image sequences.** Synthetic semantic-variation sequences are harder to construct in a controlled manner when the generator is misaligned with the target domain. In our preliminary attempts, domain-specialized generators (e.g., StyleGAN (Karras et al., 2019) and CycleGAN (Zhu et al., 2017)) often produced unrealistic outputs when asked to enforce semantic shifts on out-of-domain objects, and diffusion-based generators (e.g., InstructPix2Pix (Brooks et al., 2023)) frequently introduced visible artifacts or drifted from the input identity, which injects unintended semantic factors. We therefore use multimodal large language models (MLLM) (e.g., GPT models (Achiam et al., 2023) and Seedream (Guo et al., 2025)) to construct synthetic image sequences. Concretely, for each dataset we choose three representative classes and sample seed images per class. For each seed image, we generate at least one pair of ID and OOD semantic shifts for each semantic. Each shift is instantiated as an ordered image sequence with an explicit semantic progression (e.g., for a food class such as *samosa*, we generate shape variations from triangular to round as a controlled semantic factor). We curate generated outputs to ensure visual coherence and semantic correctness: samples with severe artifacts, identity mismatch, or off-target edits are discarded. Overall, while these synthetic sequences enable targeted ID/OOD semantic shifts, they also highlight that constructing a fully controlled semantic-variation dataset remains a challenge for robustness evaluation.

**Real-world image sequences.** For each dataset, we ensure coverage by selecting a subset of classes that accounts for at least 10% of all categories. For each selected class, we embed all images using the CLIP image encoder and perform automatic clustering in the embedding space based on pairwise distances, aiming to expose naturally occurring intra-class modes (e.g., color, shape, texture, background). We then specify a set of attribute prompts per class using prior knowledge of plausible variations (e.g., color for flowers, shape for certain foods), while keeping a shared prompt template (e.g., ``a photo of a {attribute} {class}'') to reduce confounding from prompt format. For each prompt, we rank clusters by their proximity to the prompt embedding. Finally, we manually select representative images from the top-ranked clusters to form an ordered image sequence that exhibits a gradual semantic change. This manual step is necessary because purely automatic ranking can still surface outliers or cases where the target attribute is entangled with unintended factors.

## C. Cross-Modal Validity of Language Proxies

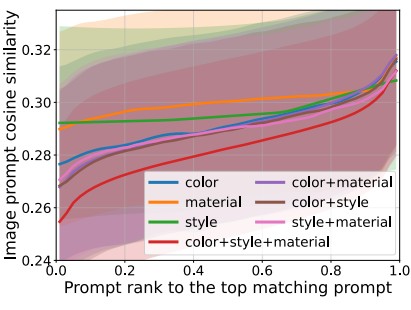

(a) Cosine similarity (image to prompts).

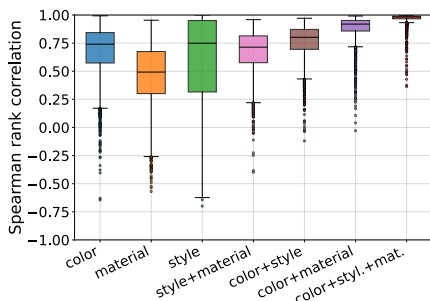

(b) Spearman correlation within concept.

*Figure 7.* Cross-modal semantic consistency on an ImageNet subset. For each image, we form a prompt family by fixing a template and varying only the attribute word, e.g., *"a photo of a [attribute] [class]"*. (a) We select the most similar prompt to the image as the anchor $t^*$, rank the remaining prompts by their cosine similarity to $t^*$ in the prompt embedding space, and plot image-to-prompt cosine similarity over the normalized text-induced rank. (b) Box plots of Spearman $\rho$ measure the agreement between the within-family prompt ranking by prompt-to-prompt cosine similarity to the anchor prompt $t^*$ and the prompt ranking by image-to-prompt cosine similarity to the image $x$.

Figure 7 evaluates cross-modal semantic consistency on an ImageNet subset using controlled semantic families (e.g., color, material, and style.) Within each semantic family, we fix a template and vary only the attribute word to form a prompt family, e.g., *"a photo of a [color] [class]"*. For each image $x$, we compute its cosine similarity to every prompt in the family and select the most similar one as the anchor prompt $t^*$. We then rank the remaining prompts by their cosine similarity to $t^*$. Figure 7(a) plots image-to-prompt cosine similarity over the normalized rank induced by this text-side ordering. Figure 7(b) reports Spearman correlation $\rho$ between the prompt ranking by prompt-to-prompt similarity to the anchor prompt $t^*$ and the prompt ranking by image-to-prompt cosine similarity to the image $x$.

Across all concept families, we observe samples with $\rho$ close to 1 in every family, indicating that the prompt ordering induced by prompt-to-prompt similarity to $t^*$ can be consistent with the ordering induced by image-to-prompt similarity. Higher correlation therefore suggests that relative similarity relations within the prompt family are frequently preserved across modalities, supporting the use of text prompts as semantic proxies in our framework. Compositional prompt families, formed by combining multiple attributes such as color and style, tend to exhibit higher $\rho$, suggesting that richer semantic descriptions improve proxy quality under the same controlled setup. A plausible explanation is that additional attributes provide more context for grounding the varied word, reducing ambiguity and making the intended semantic direction more specific. This indicates that prompt engineering (Zhou et al., 2022a;b) can serve as a practical mechanism to strengthen semantic specification when constructing language proxies. While proxy quality can vary across concepts and samples, the overall trend in Figure 7 supports our use of prompt families to parameterize semantic variation and to derive certificates based on the induced embedding geometry.

## D. Alignment to Visual Reference Transformations

In this section, we evaluate whether our constructed semantic transformation $\gamma$ in the embedding space aligns with semantic variation in the input space. Using annotated image sequences that exhibit gradual changes of the target semantic, we build a visual reference transformation $\gamma^{\text{ref}}$ and compare it with our established transformations including the text-specified (T-Spec) and image-specified (I-Spec) methods, as well as the ExactLine baseline. Specifically, we use a uniform extent grid $\Phi := \{0, \frac{1}{K-1}, \ldots, 1\} \subset [0, 1]$ and compute the cosine similarity between $\gamma(\varphi)$ and $\gamma^{\text{ref}}(\varphi)$ at each $\varphi \in \Phi$. We then average over $\Phi$ to obtain the mean cosine similarity, $\frac{1}{|\Phi|} \sum_{\varphi \in \Phi} \cos\big(\gamma(\varphi), \gamma^{\text{ref}}(\varphi)\big)$.

Figure 8 reports the distribution of mean cosine similarity scores over semantic instances for each dataset. T-Spec and I-Spec consistently achieve higher alignment than ExactLine, demonstrating that our constructed transformations closely track the embedding variation induced by annotated input sequences with gradual semantic changes. In contrast, linear interpolation in the input space can produce embedding paths that deviate substantially from the visual reference transformation. Overall, these results show that our constructed transformations can faithfully follow semantic variation as realized in the input space, providing a concrete input-space grounding for our embedding-space specification and supporting its use for certificate construction.

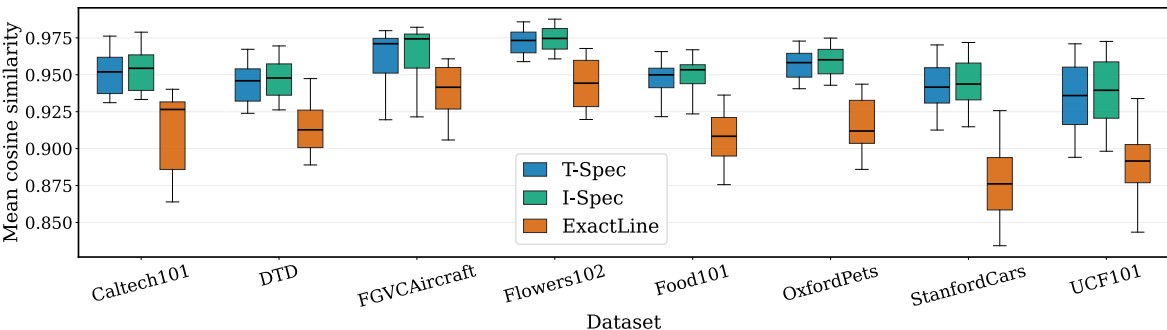

*Figure 8.* Alignment to semantic variation in the input space. For each dataset, we compare our constructed semantic transformation with a visual reference transformation constructed from the annotated image sequence with gradual semantic variations. For each semantic instance, we uniformly sample extents $\varphi \in [0, 1]$, compute cosine similarity between the transformed embedding and the visual reference embedding at each $\varphi$, and average over extents to obtain a mean cosine similarity. Boxplots show the distribution of mean cosine similarity across datasets. Both our text-specified transformation (T-Spec) and image-specified transformation (I-Spec) consistently outperform ExactLine, showing strong alignment with semantic variation in the input space.

# E. Stability under Prompt Variations

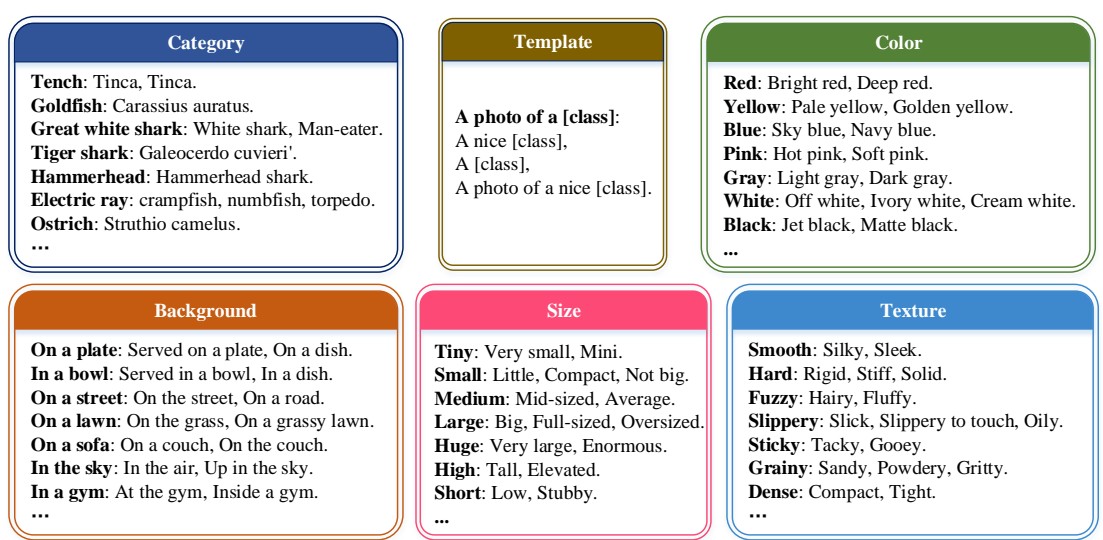

*Figure 9.* Examples of prompt variants on ImageNet. We consider category name variants from ImageNet annotations, template variants for VLM prompting, and attribute synonym sets for semantics such as color, background, size, and texture. Within each type, we form a prompt family by substituting only the corresponding word or phrase while keeping the remaining prompt structure fixed, e.g., *"a photo of a Tench"* → *"a photo of a Tinca"*.

We evaluate whether the proxy-specified semantics remain stable under typical prompt variations. We consider three types of prompt variants, as illustrated in Figure 9. These include category name variants from ImageNet annotations, commonly used prompt template variants for VLM prompting, and attribute synonym sets for semantics such as color, background, size, and texture. For each prompt variant, we vary only one prompt component at a time and measure the resulting change in image-to-prompt cosine similarity across variants.

Figure 10 reports prompt-to-prompt cosine similarity to quantify how much a prompt changes under different prompt variation types on an ImageNet subset. For each prompt family, we select a reference prompt and compute the cosine similarity between its prompt embedding and the embeddings of its variants. We aggregate these similarities across prompt families and report the resulting distributions for each variation type. The results show consistently high prompt cosine similarity for template variants and attribute synonym substitutions, indicating that these edits remain close to the reference phrasing in the prompt embedding space. Category name substitutions exhibit a larger spread. This is expected because ImageNet category annotations can include heterogeneous strings, such as scientific names, alternative spellings, and

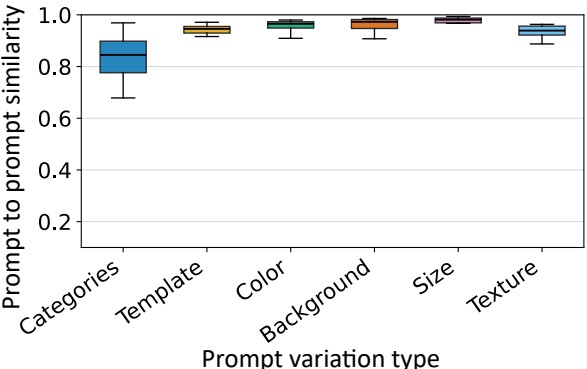

*Figure 10.* Prompt cosine similarity under prompt variations on an ImageNet subset. For each variation type, we compute cosine similarity between a reference prompt (shown in bold) and its variants within the same prompt family, and report the distribution across prompt families.

uncommon aliases. For example, a class may be annotated with both a common name and a scientific name (e.g., *goldfish* vs. *Carassius cuvieri*), or with alternative aliases (e.g., *great white shark* vs. *man-eater*). These substitutions can shift the prompt embedding more substantially than template or attribute edits. Nevertheless, the overall similarities remain high, indicating that category name variants still preserve relative image–prompt alignment for most classes.

Overall, these results demonstrate that using language as semantic proxies is robust to small prompt-level variations. This provides empirical support that the resulting transformations are not overly sensitive to the particular wording used to specify a target semantic.

## F. Semantic Strength via Similarity

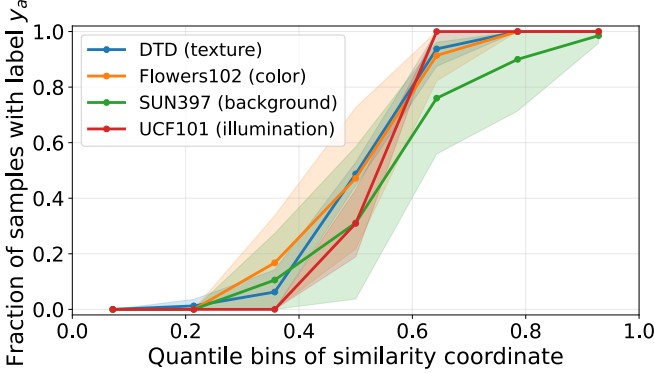

*Figure 11.* Semantic strength via similarity. For each dataset, we sample 20 semantic pairs $(a, a')$ and randomly split images from the two classes into two disjoint subsets. We compute class mean visual embeddings $\bar{z}_a, \bar{z}_{a'}$ on images from one subset and form the semantic direction by $v_{a,a'} = \bar{z}_{a'} - \bar{z}_a$. We then score images in the other subset by $t(x_i) = \langle z_i, v_{a,a'} \rangle$ with $z_i = f_{\text{img}}(x_i)$, sort by $t(x_i)$, and partition into equal-count quantile bins. The plot shows the fraction of samples with label $y_{a'}$ across bins. Solid lines are averages over pairs and shaded bands are $95\%$ normal approximation confidence intervals across pairs.

Our certificates rely on the assumption that semantic strength can be measured by similarity in the embedding space. While we derive bounds that account for semantic misalignment, we also provide empirical evidence that tests whether samples from two semantic instances can be consistently ordered by a single similarity coordinate in VLMs. We consider four datasets that represent common semantic factors. DTD focuses on texture attributes (e.g., *dotted*, *knitted*, and *woven*), making it a natural choice for texture variation. Flowers102 contains flower categories with stable color appearance within each category, making it suitable for testing color variation. SUN397 spans a broad range of scene types, and class differences often manifest through global background context and layout, for example *airplane cabin* versus *desert*. UCF101 consists of video frames collected under diverse capture conditions, where lighting and viewpoint changes can be prominent, and we use it as a practical proxy for illumination related variation. Together, these datasets allow us to test similarity-based

semantic strength across heterogeneous semantic factors.

For each dataset, we instantiate semantic variations by sampling 20 semantic pairs $(a, a')$, where $a$ and $a'$, instantiated by two classes in the dataset. For each pair $(a, a')$, we collect images from the two classes and randomly split them into two disjoint subsets. For an image $x_i$, we denote its visual embedding by $z_i = f_{\text{img}}(x_i)$, and denote its label within the pair by $y_i \in \{y_a, y_{a'}\}$. On one subset, we compute class mean embeddings $\bar{z}_a$ and $\bar{z}_{a'}$ by averaging $z_i$ within each class, and define the semantic direction $v_{a,a'} = \bar{z}_{a'} - \bar{z}_a$. On the other subset used for evaluation, we compute a similarity coordinate for each image as $t(x_i) = \langle z_i, v_{a,a'} \rangle$. We pool the images in the evaluation subset from both classes, sort them by $t(x_i)$, and partition them into multiple equal-count quantile bins. For each bin $\mathcal{B}$, we compute the fraction of samples labeled $y_{a'}$ as $\frac{1}{|\mathcal{B}|} \sum_{x_i \in \mathcal{B}} \mathbb{I}[y_i = y_{a'}]$, where $y_i$ denotes the label of $x_i$ within the sampled pair. Repeating this procedure over the sampled pairs yields one sequence of fractions per pair. Figure 11 aggregates these results for each dataset by plotting the average across pairs within each bin as the solid line, with a shaded band showing a 95% normal approximation confidence interval for this average based on the empirical variability across pairs.

Figure 11 shows that the fraction of samples with label $y_{a'}$ increases across the quantile bins of the similarity coordinate on all four datasets. The transition is more concentrated for DTD and Flowers102 and more gradual for SUN397, which is expected given the larger intra class variation in scene categories. Overall, the results indicate that semantic strength is captured by similarity in the embedding space across different semantic factors.

