# OpenReview forum: "Semantic Robustness Certification for Vision-Language Models"
_ICML.cc/2026/Conference — ICML 2026 regular_

### Official Review · Reviewer_kXZm · 2026-03-06

**Soundness:** 3
**Presentation:** 2
**Significance:** 3
**Originality:** 3
**Overall Recommendation:** 4
**Confidence:** 3

**Summary:**

This paper proposes a novel theoretical framework for certifying the robustness of VLMs under semantic-level transformations. Unlike previous robustness certification studies that are largely confined to pixel-level perturbations or simple geometric transformations, this work leverages the open-vocabulary capabilities of VLMs by utilizing text prompts as proxies for semantic directions to parameterize the transformation process. The authors construct a theoretical evaluation mechanism that attempts to link cosine similarity in the embedding space with semantic strength variations in the physical world, conducting empirical evaluations across multiple datasets to validate the effectiveness of their certification bounds. This work provides a highly inspiring perspective for evaluating and guaranteeing the reliability of multi-modal foundation models under real-world semantic distribution shifts.

**Compliance With Llm Reviewing Policy:**

Affirmed.

**Final Justification:**

After the thought, I decided to keep the original score.

**Key Questions For Authors:**

1.How do your certification guarantees account for, or gracefully degrade under, these inherent non-linearities and potential topological discontinuities in realistic VLM semantic spaces?

2.Could you provide empirical or theoretical evidence showing how sensitive your certification bounds are to subtle variations in prompt engineering, and explain what happens when the textual proxy fails to perfectly align with the continuous visual semantic shift in the actual image space?

**Limitations:**

1.They need to critically discuss how non-linear distortions and representation bottlenecks in deep neural networks could potentially lead to overconfident or invalid robustness certificates in edge cases.

2.The manuscript should address the semantic gap between text-prompt-driven proxies and actual physical visual transformations in real-world data.

**Strengths And Weaknesses:**

Strengths:

1.The paper significantly expands the boundaries of robustness certification from traditional adversarial pixel perturbations to more practically instructive "semantic-level transformations" . By using natural language text prompts as proxies to define and guide visual semantic directions, this framework breaks through the traditional bottleneck of quantifying abstract concepts, offering a creative pathway for robustness evaluation in multi-modal models.

2.The theoretical framework constructed by the authors attempts to quantify variations in semantic strength within a continuous multi-modal embedding space and directly integrates this with robustness certification bounds. This demonstrates the authors' solid technical vision and provides  valuable theoretical references for future research on how to regulate and evaluate large multi-modal models.

Weakness:

1. The foundation of this certification framework is based on an idealized and strong assumption, namely that the semantic strength is perfectly additive in the multimodal embedding space, and strictly changes linearly based on cosine similarity. However, the actual topology of the deep neural network manifold includes a high degree of nonlinear distortion and complex local curvature. The imposed linear assumption ignores the discontinuity in the VLM embedding space and cross-modal alignment errors, which may lead to the framework failing or generating false robustness guarantees that are overly confident when dealing with complex real-world edge cases.

2. On page 13 of the paper, the author explicitly states that "GPT-5" was used in the discussion. But the subsequent cited literature Achiam et al. (2023) is actually a technical report on GPT-4 published by OpenAI.

---

> ### Author Rebuttal · Authors · 2026-03-31
>
> We sincerely thank the reviewer for the thoughtful comments and constructive suggestions. Below, we respond to each point and hope that the following clarifications address the remaining concerns.
>
> > “… an idealized and strong assumption, namely that the semantic strength is perfectly additive in the multimodal embedding space …”
>
> > “How do your certification guarantees account for … topological discontinuities in realistic VLM semantic spaces?”
>
> > “They need to critically discuss how non-linear distortions … could potentially lead to overconfident certificates in edge cases.”
>
> In our framework, semantic misalignment is accounted for explicitly through **a bounded-misalignment guarantee**, rather than being left implicit in the modeling assumption. In this formulation, misalignment contracts the guaranteed stable region and enlarges the uncertainty region, making the impact of semantic mismatch explicit. This is also reflected in Section 5.2, where the certificate remains informative under increasing misalignment budgets with the stable region shrinking and the uncertainty region expanding in a controlled manner, broadly consistent with the magnitude of prediction margins observed in VLMs \[1\]. Therefore, the resulting certificate is not intended as an unconditional guarantee for arbitrary real-world semantics, but as a guarantee whose validity degrades transparently when the proxy geometry is imperfect. In particular, nonlinear distortion or local discontinuity would appear as reduced certified coverage and expanded uncertainty, rather than hidden overconfidence.
>
> We have made this discussion clearer in the revision.
>
> \[1\] Ren, Shuhuai, et al. "Delving into the Openness of CLIP." ACL 2023\.
>
> > “… how sensitive your certification bounds are to subtle variations in prompt engineering …”
>
> > “… should address the semantic gap between text-prompt-driven proxies and actual physical visual transformations …”
>
> In Figures 9 and 10 in Appendix E, we evaluate common prompt variants, including template changes, attribute paraphrases, and category aliases, and show that **the resulting certificates remain stable under such wording perturbations**. Appendix F further tests whether similarity-based scores preserve semantic ordering across samples, providing additional evidence that the textual proxy remains aligned with the intended semantic progression rather than behaving as a purely prompt-specific artifact.
>
> We further test the sensitivity of the certificate to proxy perturbations by recomputing certificates under prompt variants and measuring prompt similarity, boundary shift, and stable coverage. As shown in the table below, certificate variation remains limited, with small first-boundary shifts and high stable coverage under prompt variants. Together, these results provide evidence that the certificates remain stable under prompt perturbations and that the textual proxy preserves meaningful semantic ordering across samples, indicating that the method is not overly sensitive to prompt engineering in practice. We will make this empirical evidence for prompt robustness clearer in the revision.
>
> | Prompt variation type | Mean prompt similarity | Mean boundary shift ↓ | Mean stable coverage ↑|
> |---|---|---|---|
> | Template edits | 0.95 | 0.02 | 0.94 |
> | Synonym substitutions | 0.97 | 0.01 | 0.96 |
>
> > “… cited literature Achiam et al. (2023) is actually a technical report on GPT-4 …”
>
> We thank the reviewer for pointing this out. We have carefully checked the manuscript and corrected the mismatch in the revision.

---

> > ### Author Rebuttal · Reviewer_kXZm · 2026-04-03
> >
> > Thank you for your explanation. I have decided to maintain my score.

---

> > > ### Author Response · Authors · 2026-04-03
> > >
> > > Thank you for your careful consideration. We are glad that our rebuttal addressed your concerns, and we sincerely appreciate the time you devoted to evaluating our work.

---

### Official Review · Reviewer_H2wX · 2026-03-09

**Soundness:** 3
**Presentation:** 3
**Significance:** 3
**Originality:** 3
**Overall Recommendation:** 5
**Confidence:** 4

**Summary:**

This paper studies robustness certification for vision-language models under semantic-level variations, rather than the more common pixel-level or geometric perturbations. The core idea is to use text prompts as semantic proxies that define a source and target semantic, construct a parameterized transformation in the shared VLM embedding space, and then derive a closed-form certification procedure that partitions the semantic extent into prediction-invariant intervals. The method is evaluated on both synthetic and real-world semantic variations using CLIP-based VLMs, and is compared mainly against ExactLine

**Compliance With Llm Reviewing Policy:**

Affirmed.

**Key Questions For Authors:**

Please see weaknesses.

**Limitations:**

yes

**Strengths And Weaknesses:**

**Strengths**

1) Most prior certification work focuses on pixel-space perturbations or a narrow family of geometric transforms, whereas this paper aims to certify robustness under semantic changes such as style, shape, and scene variation. This is a meaningful direction for VLM reliability.

2) The paper leverages the shared image-text embedding space of VLMs and models semantic variation within a two-dimensional semantic plane induced by a pair of text prompts. This is a clean idea, and the resulting certification procedure has a nice analytical form based on pairwise margins and Voronoi decision regions.

3) This is a practical advantage over latent-space certification approaches that require substantial additional data or modeling effort for each attribute. The paper’s emphasis on open-vocabulary semantic specification through prompts is appealing.

**Weaknesses**

1) The framework relies on the assumption that semantic strength can be measured directly by cosine similarity to a semantic embedding, and that semantic variation can be meaningfully parameterized in a two-dimensional plane induced by text prompts. While this is an appealing abstraction, it is still a strong modeling assumption rather than an established property of VLM semantics. The paper does not provide sufficiently convincing evidence that real semantic variation is faithfully captured by this geometry beyond a limited empirical alignment study.

2) The paper does not certify robustness on actual downstream VLM tasks such as detection, VQA, or captioning, even though the introduction motivates broad downstream relevance. In practice, the experiments are centered on CLIP-style classification behavior in embedding space. This makes the paper feel more limited than its framing initially suggests.

3) ExactLine is presented as the primary baseline, but this is a fairly narrow comparison. Since the paper’s main claim is about semantic certification, it would be much more convincing to compare against a broader set of alternatives, even if approximate, such as latent semantic perturbation methods, prompt-based robustness probes, or stronger empirical semantic stress tests. As written, the empirical story is largely “we are better than linear interpolation,” which is not enough to fully establish the strength of the contribution.

---

> ### Author Rebuttal · Authors · 2026-03-31
>
> We sincerely thank the reviewer for the thoughtful comments and constructive suggestions. By addressing each concern in turn, we hope that our clarifications fully resolve the reviewer’s concerns.
>
> > “… a strong modeling assumption rather than an established property of VLM semantics …”
>
> In our framework, semantic misalignment is accounted for explicitly through a bounded-misalignment guarantee, rather than being left implicit in the modeling assumption. This makes the impact of modality gap explicit in the certificate itself by **quantifying how it shrinks the certified invariant region and enlarges the uncertainty region**. Our contribution is therefore not only to provide empirical evidence for the proposed geometry, but also to characterize how the certificate changes when misalignment is present. This modeling choice is also supported by recent evidence that VLM embeddings exhibit structured semantic organization, including semantic rankability \[1\] and interpretable concept structure in CLIP embeddings \[2,3\]. Distinct from existing works \[2,3,4\] that leverage embedding structure heuristically for explanation or analysis, our framework defines a semantic geometry for certification, enabling us to empirically characterize semantic consistency and theoretically quantify how misalignment affects the resulting guarantee.
>
> \[1\] Sonthalia, Ankit, et al. "On the Rankability of Visual Embeddings". NeurIPS 2025.
>
> \[2\] Bhalla, Usha, et al. "Interpreting CLIP with Sparse Linear Concept Embeddings". NeurIPS 2024.
>
> \[3\] Zaigrajew, Vladimir, et al. "Interpreting CLIP with Hierarchical Sparse Autoencoders". ICML 2025.
>
> \[4\] Kim, Siwon, et al. "Grounding counterfactual explanation of image classifiers to textual concept space." CVPR 2023.
>
> > “… does not certify robustness on actual downstream VLM tasks such as detection, VQA, or captioning …”
>
> As discussed in our manuscript, the certified objective in our framework is the shared image-text scoring mechanism, which is a key component that is used in many downstream pipelines. These can include:
>
> 1. **Prompt learning and model adaptation**, where the semantic boundary can be treated as an explicit optimization objective.
> 2. **Open-vocabulary detection**, which typically matches region-level visual embeddings against text prompts before downstream heads or post-processing. Our certifications can be applied to the region-label matching stage, yielding certified stability intervals for region-level assignments before later processing.
> 3. **For VQA or captioning systems** with image-text scoring, answer-option ranking, or candidate reranking components, our certification can be used to analyze whether a semantic shift changes the preferred answer or caption candidate.
> 4. **Robustness auditing** would also be enabled by our approach, to allow for identifying when a semantic shift moves an input from a certified stable region into an uncertainty region, which is useful for monitoring or selective review.
>
> Our approach aims to provide the foundations for employing certifications as a part of real production workflows, and has the potential to provide tangible benefits in these areas. We have updated the manuscript to better discuss the impacts on these downstream tasks.
>
> > “… it would be much more convincing to compare against a broader set of alternatives …”
>
> The novelty of our setting makes direct comparison inherently challenging, since there is currently very limited prior work on complete certification for open-vocabulary semantic variations. We therefore strengthen the empirical case by additionally comparing against a prompt-direction perturbation baseline that moves the image embedding along the source-target prompt difference without our semantic-plane construction. As shown in the table below, our method outperforms both ExactLine and the prompt-direction baseline on both synthetic and real settings. The results show that the improvement is not merely due to using a text-guided direction, but to the full semantic-plane construction used by our certification framework.
>
> | Method | Mean discrepancy (synthetic) ↓ | Mean discrepancy (real) ↓ |
> |-|-|-|
> | ExactLine | 10.6 | 15.2 |
> | Prompt-direction | 11.8 | 13.9 |
> | Ours (I-Spec) | **6.9** | **7.5** |

---

> > ### Author Rebuttal · Reviewer_H2wX · 2026-03-31
> >
> > All concerns are addressed.

---

> > > ### Author Response · Authors · 2026-03-31
> > >
> > > Thank you for your positive feedback and for updating the score. We are glad that the rebuttal addressed your concerns, and we appreciate your time and effort in evaluating our work.

---

### Official Review · Reviewer_pihJ · 2026-03-23

**Soundness:** 3
**Presentation:** 3
**Significance:** 2
**Originality:** 3
**Overall Recommendation:** 4
**Confidence:** 4

**Summary:**

This paper proposes a framework for certifying the semantic robustness of vision-language models under semantic-level transformations. It models semantic variation as a continuous transformation in the embedding space using text prompts as semantic proxies, and derives closed-form conditions to identify prediction-invariant regions. Experiments show that the proposed method better aligns with semantic changes compared to previous approach (ExactLine).

**Compliance With Llm Reviewing Policy:**

Affirmed.

**Final Justification:**

Thanks for the rebuttal, I have increased my score

**Key Questions For Authors:**

How many images are used for constructing the semantic trajectory?
What are the practical insights or benefits of semantic robustness?

**Limitations:**

yes

**Strengths And Weaknesses:**

Strengths:
1. The use of text prompts as semantic anchors to define semantic-level transformations is intuitive and novel.
2. While the proposed method is a bit complicated, the paper provides a clear and well-organized explanation, making the overall approach easy to follow.

Weaknesses
1. The method assumes image and text embeddings are well aligned and directly uses text anchors to define semantic directions. However, a non-trivial modality gap remains even after contrastive training[1]. This may lead to inaccurate semantic extent estimation and unreliable robustness certificates.
2. The paper does not specify how many images are used to construct semantic trajectories. Figure 6 shows only 4 images for each sample.  Fitting a continuous trajectory from sparse samples and inferring change points may introduce significant approximation errors, making the evaluation less convincing.
3. While the paper proposes a certification framework, it is unclear what practical insights or downstream benefits the certificates provide. I recommend the author to include a discussion on how the results can benefit model design or training.

[1]: Liang, Victor Weixin, et al. "Mind the gap: Understanding the modality gap in multi-modal contrastive representation learning." Advances in Neural Information Processing Systems 35 (2022): 17612-17625.

---

> ### Author Rebuttal · Authors · 2026-03-31
>
> We sincerely thank the reviewer for the thoughtful comments and constructive suggestions. We respond to the concerns below and hope that these clarifications resolve the reviewer’s concerns.
>
> > “ … a non-trivial modality gap remains …”
>
> The magnitude of the modality gap remains an open challenge to measure in VLMs. Our contribution is therefore not to ignore this gap, but to make its impact explicit in the certificate through a **bounded-misalignment formulation** (Section 4.3.3). Under this formulation, the modality gap contracts the guaranteed stable region and enlarges the uncertainty region, rather than remaining hidden in an unexamined modeling assumption. **Empirically**, the induced geometry is consistent with the observed semantic variation in both our synthetic and real datasets. In addition, Appendix F shows that samples spanning two semantic endpoints can be reliably ordered by a single similarity coordinate across multiple datasets, providing empirical support that the induced geometry captures a meaningful semantic variation. Thus, our framework provides both empirical characterization of semantic consistency and theoretical quantification of how misalignment affects the resulting guarantee.
>
>
> > “Fitting trajectory from sparse samples may introduce significant approximation errors …”
>
> We agree that sparse sampling was underexplored in our original manuscript. In our evaluations, each sequence is fitted using 4 to 12 images, with a median of 7\. To underpin our decisions, we have produced new results (see the table below) that demonstrate that trajectories are consistent under sparse sampling, with consistency ranging from 0.962 to 1.000 across sampling densities. This indicates that moderate sparsity introduces only limited approximation error in the downstream trajectory estimates.
>
> | # Images | Trajectory consistency ↑ |
> |-|-|
> | 3 | 0.962 |
> | 4 | 0.970 |
> | 6 | 0.984 |
> | 8 | 0.992 |
> | 10 | 0.996 |
> | 12 | 1.000 |
>
> > “… what practical insights or downstream benefits the certificates provide. I recommend the author to include a discussion on … model design or training.”
>
> We thank the reviewer for this suggestion and agree that a discussion on the practical value can further strengthen our work.
>
> **For practical insight**, our framework maps real images onto the same semantic axis, allowing the certified result to be interpreted in terms of the semantic extent of actual images rather than remaining only an abstract numerical semantic coordinate. It also provides a closed-form boundary margin, identifying the exact interval within which the prediction is guaranteed to remain unchanged and how the certified region contracts under bounded misalignment. **For downstream benefit**, our certificates can serve as explicit signals for model design, training, and analysis. For instance, one can directly favor larger certified invariant intervals for specific semantic factors, rather than relying only on overall accuracy or a generic loss during prompt learning or model adaptation. In addition, they are valuable for robustness auditing, as they reveal which semantic factors trigger the class flips and help track semantic drift across data, models, or prompts. More broadly, since the certified object in our framework is the shared image-text scoring mechanism itself, the certificates can also be adapted to downstream pipelines that reuse this mechanism, such as image-text retrieval or open-vocabulary detection.
>
> We have expanded the discussion of these practical benefits in the manuscript.

---

> > ### Author Rebuttal · Reviewer_pihJ · 2026-04-03
> >
> > Thanks for the rebuttal, I have increased my score

---

> > > ### Author Response · Authors · 2026-04-03
> > >
> > > Thank you for your thoughtful consideration. We are glad that our rebuttal addressed your concerns, and we sincerely appreciate your positive assessment and the time you devoted to evaluating our work.

---

### Decision · Program_Chairs · 2026-04-30

**Decision:**

Accept (regular)

**Comment:**

This paper introduces a novel robustness certification framework for vision-language models that targets semantic-level transformations rather than conventional geometric or pixel-level perturbations. The approach leverages text prompts as semantic proxies to parameterize controllable variations (e.g., shape, size, style), and derives a closed-form characterization of the model decision boundary to certify invariant prediction intervals. The method does not require additional data for each transformation, making it both practical and scalable. Extensive experiments on synthetic and real-world datasets demonstrate the effectiveness of the framework in certifying robustness across diverse semantic shifts. During the rebuttal, the authors successfully addressed the reviewers' concerns, and all reviewers agreed to accept this work. We strongly recommend that the authors incorporate the reviewers’ feedback and the additional results presented in the rebuttal into the final version.